# Sporadic Amyotrophic Lateral Sclerosis Skeletal Muscle Transcriptome Analysis: A Comprehensive Examination of Differentially Expressed Genes

**DOI:** 10.3390/biom14030377

**Published:** 2024-03-20

**Authors:** Elisa Gascón, Pilar Zaragoza, Ana Cristina Calvo, Rosario Osta

**Affiliations:** 1Department of Anatomy, Embryology and Animal Genetics, University of Zaragoza, 50013 Zaragoza, Spain; egascon@unizar.es (E.G.); pilarzar@unizar.es (P.Z.); accalvo@unizar.es (A.C.C.); 2Centro de Investigación Biomédica en Red de Enfermedades Neurodegenerativas (CIBERNED), 28029 Madrid, Spain; 3Agroalimentary Institute of Aragon (IA2), University of Zaragoza, 50013 Zaragoza, Spain; 4Institute of Health Research of Aragon (IIS), 50009 Zaragoza, Spain

**Keywords:** neurodegenerative diseases (NDDs), extracellular/circulating biomarkers, microRNA, amyotrophic lateral sclerosis (ALS), bioinformatics

## Abstract

Amyotrophic lateral sclerosis (ALS) that comprises sporadic (sALS) and familial (fALS) cases, is a devastating neurodegenerative disorder characterized by progressive degeneration of motor neurons, leading to muscle atrophy and various clinical manifestations. However, the complex underlying mechanisms affecting this disease are not yet known. On the other hand, there is also no good prognosis of the disease due to the lack of biomarkers and therapeutic targets. Therefore, in this study, by means of bioinformatics analysis, sALS-affected muscle tissue was analyzed using the GEO GSE41414 dataset, identifying 397 differentially expressed genes (DEGs). Functional analysis revealed 320 up-regulated DEGs associated with muscle development and 77 down-regulated DEGs linked to energy metabolism. Protein–protein interaction network analysis identified 20 hub genes, including *EIF4A1*, *HNRNPR* and *NDUFA4*. Furthermore, miRNA target gene networks revealed 17 miRNAs linked to hub genes, with hsa-mir-206, hsa-mir-133b and hsa-mir-100-5p having been previously implicated in ALS. This study presents new potential biomarkers and therapeutic targets for ALS by correlating the information obtained with a comprehensive literature review, providing new potential targets to study their role in ALS.

## 1. Introduction

Amyotrophic lateral sclerosis (ALS) is a devastating neurodegenerative disorder, characterized by the progressive degeneration of upper and lower motor neurons, resulting in muscle atrophy and diverse clinical manifestations. The disease exhibits an estimated prevalence of 4–8 cases per 100,000 individuals across most populations [1]. Alongside motor neuron degeneration, ALS patients may present cognitive and behavioral impairments, adding to the complexity of the disease [2,3]. The median survival post-symptom onset typically ranges from 3 to 5 years, although prognostic factors can influence this timeframe [4].

ALS is broadly classified into sporadic (sALS) and familial (fALS) forms, accounting for 90–95% and 5–10% of cases, respectively [5]. Presently, only two approved drugs, edaravone and riluzole, are available for ALS treatment, emphasizing the urgent need for further therapeutic options [6].

While the precise etiology of ALS remains elusive, genetic factors related to RNA metabolism, protein homeostasis, DNA damage repair, nucleocytoplasmic transport, excitotoxicity, oxidative stress and axonal transport have been implicated [7,8,9,10,11,12,13]. Specific mutations have been described in genes such as *SOD1*, *FUS*, *C9ORF72*, *ATXN2*, *OPTN*, *VCP*, *PFN1*, *MATR3*, *SETX* and *UBQLN2* [14,15,16,17,18,19,20,21,22,23,24]. Advancements in whole-genome sequencing have led to the discovery of new ALS-associated genes such as *LGALSL*, *FIG4* or *ALS2* [25,26,27]. Despite these insights, ALS remains a multifactorial disease, which reflects the difficulty of achieving more effective therapies [28].

Biomarkers can play a pivotal role in ALS research, enabling early diagnosis, prognosis prediction, treatment evaluation and therapeutic discovery [29]. Bioinformatics, a field of remarkable progress, has become relevant in the biomarker exploration of various diseases [30,31,32]. Previous studies conducted by Lin, Huang, Chen, Ye, Su and Yao [2] have delved into bioinformatic analyses using gene expression series (GSE) from human spinal cord motor neuron data. However, human muscle samples, a primary site affected by ALS, remain largely unexplored. Interestingly, growing evidence suggests alterations in the neuromuscular junction from the presymptomatic stage. These alterations could indicate that the degenerations observed in motor neurons could be influenced by pathogenic alterations in muscles [33].

In this study, we scrutinized the GEO dataset GSE41414 [34] to identify differentially expressed genes (DEGs) in sALS-affected muscle samples compared to healthy controls. Subsequently, GO and pathway enrichment analyses shed light on the functions and pathways influenced by these DEGs. Furthermore, we constructed protein–protein interaction networks to pinpoint subnetworks and hub genes. Additionally, miRNA–target gene networks were developed, offering insights into potential gene interactions in the context of ALS.

## 2. Materials and Methods

### 2.1. Microarray Dataset

An extensive search was conducted within the Gene Expression Omnibus (GEO) database, a globally accessible public repository established in 2020 to investigate pertinent gene expression datasets [35]. The search strategy used a combination of specific keywords: ALS and skeletal muscle (ALS [All Fields] AND skeletal muscle [All Fields]). Stringent criteria were applied, including a restriction to datasets exclusively associated with *Homo sapiens* and categorized under the study type of expression profiling by array. We focused on samples of skeletal muscle to provide new insights on this tissue that is affected by the disease.

Among the identified datasets, GSE41414 emerged as the dataset of choice. This dataset, residing on the Affymetrix Human HG-Focus Target Array platform, encompasses a comprehensive collection of samples, including seven control and seven sporadic ALS (sALS) patient specimens (Appendix A). All control samples encompass fibers from the deltoid skeletal muscle. For samples from individuals with sporadic amyotrophic lateral sclerosis (sALS), three come from the quadriceps skeletal muscle, and four come from the deltoid skeletal muscle. Both cohorts underwent extensive analysis, and subsequent observation revealed no discernible differences. Consequently, these groups were combined into a singular category representing skeletal muscle affected by sporadic amyotrophic lateral sclerosis (sALS).

### 2.2. Identification and Analysis of Differentially Expressed Genes (DEGs)

The gene expression data were analyzed using RStudio environment and specific Bioconductor packages [36]: affy (v1.78.2), oligo (v1.64.1), GEOquery (v2.68.0), limma (v3.56.2) and ggplot2 (v3.4.3). Data correction and normalization were initially performed. Subsequently, the limma package’s moderated *t*-test, based on the empirical parametric Bayes method, identified differentially expressed genes (DEGs) between ALS patient and control samples. The criteria for DEGs included |logFC| > 0.5 (1.4-fold change) for up-regulated genes, |logFC| < 0.5 (0.7-fold change) for down-regulated genes and a *p*-value < 0.05. The results were visualized using a volcano plot created with ggplot2 packages.

### 2.3. Functional and Enrichment Analysis of DEG Pathways

The functional enrichment analysis of up- and down-regulated genes was carried out using the Bioconductor R package clusterProfiler (v4.8.2) [37] with default statistical thresholds and OrgDb set to “org.Hs.eg.db”. clusterProfiler is a well-known package for performing comprehensive functional and pathway enrichment analyses that are needed for the analysis and visualization of enrichment across numerous organisms. The analysis specifically focused on Gene Ontology (GO) terms, categorizing them into (1) biological processes, (2) molecular functions and (3) cellular components. Significance was determined with a stringent criterion: GO scores with a *p*-value < 0.05 were considered statistically significant.

### 2.4. Protein–Protein Interaction (PPI) Network Construction and Subnetwork Identification

The STRING online database (https://string-db.org/, accessed on 5 February 2024) was used to predict and analyze protein–protein interactions of positively and negatively regulated genes [38]. These interactions were visually represented using Cytoscape software (v3.9.1), which allowed modification and visualization of biological networks [39]. In addition, the MCODE (Molecular Complex Detection) add-on of Cytoscape [40] facilitated the analysis of densely connected clusters within the networks based on specific criteria (degree limit = 2, node score limit = 0.2, kernel K = 2 and max. depth = 100). Subsequently, the highest scoring subnetworks for positively and negatively regulated genes were selected. For further analysis and enrichment, we used Metascape (https://metascape.org, accessed on 5 February 2024), a user-friendly online bioinformatics portal recognized for its functional enrichment and interactome analysis capabilities, which ensures meticulous exploration of the biological processes studied [41].

### 2.5. Analysis of Hub Genes and PPI Networks

To identify positively regulated and negatively regulated hub genes, the Cytoscape add-on *cytoHubba* was used. *cytoHubba* is capable of performing topological analysis using 11 methods, among which the most commonly used are degree MCC (maximum clique centrality) and betweenness [42]. The ten hub genes with the highest degrees were identified in the PPI networks that were positively and negatively regulated.

### 2.6. Prediction of miRNAs Targeting Hub Genes

The miRNet database (https://www.mirnet.ca/, accessed on 5 February 2024) served as a crucial bioinformatics platform for predicting target-gene and miRNA pairs [43]. This powerful tool integrates data from 14 distinct miRNA databases, including TarBase, miRTarBase, miRecords, miRanda, miR2Disease, HMDD, PhenomiR, SM2miR, PharmacomiR, EpimiR, starBase, TransmiR, ADmiRE and TAM 2.0. Specifically, for this research, miRNAs specific to both positively and negatively regulated hub genes were predicted. The resulting target-gene–miRNA regulatory network was depicted and visually represented using Cytoscape.

## 3. Results

### 3.1. Identification and Analysis of Differentially Expressed Genes (DEGs)

The GSE41414 dataset contained a total of 8793 genes, of which, a total of 397 differentially expressed genes (DEGs) were identified. Among these DEGs, there were 320 up-regulated genes and 77 down-regulated genes. These genes were represented in a volcano plot (Figure 1).

### 3.2. Functional and Enrichment Analysis of DEG Pathways

GO analysis consists of three parts: (1) biological processes (BPs), (2) cellular component (CC) and (3) molecular function (MF). In this study, a functional analysis of up-regulated DEGs and down-regulated DEGs (Figure 2) was performed.

The analysis revealed that up-regulated DEGs were involved in biological processes such as muscle structure development, myoblast differentiation and regulation of myoblast differentiation, among others. In terms of the CC, DEGs were mainly enriched in collagen-containing extracellular matrix, external encapsulating structure and extracellular matrix. The molecular functions associated with these DEGs were mainly structural molecule activity, mRNA binding and extracellular matrix structural constituent.

On the other hand, down-regulated DEGs were involved in biological processes of generation of precursor metabolites and energy, aerobic respiration, cellular respiration and oxidative phosphorylation, among others. The most enriched CC terms for these DEGs were the mitochondrial membrane, mitochondrial protein-containing complex and inner mitochondrial membrane protein complex. The molecular functions associated with these DEGs were electron transfer activity, oxidoreduction-driven active transmembrane transporter activity and primary active transmembrane transporter activity.

### 3.3. Protein–Protein Interaction (PPI) Network Construction and Subnetwork Identification

The protein–protein interaction network of the 397 DEGs was constructed with medium confidence using STRING. A PPI network was constructed for the up- and down-regulated DEGs. The obtained files were subsequently visualized by Cytoscape, and their possible subnetworks were analyzed with MCODE to investigate the molecular networks related to these deregulated genes. Up to nine subnetworks were identified for up-regulated DEGs and three subnetworks for down-regulated DEGs. We selected the subnetworks with the highest MCODE scores for up- and down-regulated DEGs. The subnetwork of up-regulated DEGs consisted of 15 genes: *EIF4A1*, *CCT2*, *ETF1*, *PABPC1*, *HNRNPR*, *EIF3A*, *EEF2*, *HNRNPA1*, *RPLP0*, *EEF1A1*, *RAN*, *RPL12*, *CCT6A*, *RPL15* and *CCT3* (Figure 3a). The subnetwork of down-regulated DEGs consisted of 12 genes: *COX5B*, *COX6A2*, *NDUFA4*, *COX6C*, *NDUFB4*, *ATP5MC1*, *COX8A*, *NDUFA3*, *ATP5PF*, *COX7A1*, *COX5A* and *NDUFA6* (Figure 3b).

Finally, these subnetworks were enriched using Metascape. Metascape serves as a user-friendly online bioinformatics portal, facilitating the expeditious and precise acquisition of functional enrichment and interactome analysis outcomes from a designated list of genes of interest. The most enriched pathways for the up-regulated DEG subnetwork included translation, translation factors and metabolism of RNA (Figure 4). Conversely, the most enriched pathways for the down-regulated DEG subnetwork comprised the electron transport chain, oxidative phosphorylation and respiratory electron transport (Figure 5).

### 3.4. Identification of Hub Genes

The 10 hub genes of the PPI network of up-regulated DEGs with the highest MCC (maximum clique centrality) hub according to the *cytoHubba* complement were *EEF1A1*, *RPLP0*, *EEF2*, *EIF4A1*, *CCT2*, *HNRNPR*, *RPL12*, *RPL15*, *HNRNPA1* and *PABPC1* (Figure 6a). On the other hand, the 10 hub genes in the PPI network of down-regulated DEGs with the highest MCC hub were *NDUFA4*, *COX5B*, *COX6C*, *NDUFA3*, *NDUFB4*, *COX5A*, *COX6A2*, *NDUFA6*, *COX8A* and *ATP5MC1* (Figure 6b).

By integrating the information from the 20 hub genes identified through PPI network analysis of both up- and down-regulated DEGs with the results from subnetwork identification using MCODE, it was observed that all up- and down-regulated hub genes were present in their respective highest-scoring subnetworks, which were mentioned in the previous section. This observation suggests that these 20 hub genes might serve as potential biomarkers and could lead to the identification of novel targets for amyotrophic lateral sclerosis (ALS) therapeutics. The hub genes identified as up-regulated were intricately linked to the maintenance, regeneration and differentiation of muscle tissue, as well as RNA metabolism. These findings highlighted that the maintenance of the regenerative capacity of this tissue under neurodegenerative conditions. Conversely, the down-regulated hub genes were associated with metabolism and oxidative stress, which become altered during disease progression.

Consequently, a literature review was conducted, limited to studies published within the last 5 years, to determine whether research has been conducted on each of these ALS-related genes. Among the hub genes derived from the up-regulated DEGs, no ALS-related literature was found for the *EEF1A1*, *RPLP0*, *EEF2*, *CCT2*, *RPL12* and *RPL15* genes. However, studies related to ALS were identified for the *EIF4A1* [44], *HNRNPR* [45], *HNRNPA1* [45,46] and *PABPC1* genes [47]. In the case of hub genes obtained from the down-regulated DEGs, only the *NDUFA4* gene was associated with ALS-related studies [48].

Additionally, while no studies directly related to ALS were discovered, other studies related to other neurological diseases were found in the cases of the *EEF1A1* [49], *COX5A* [50] and *RPL12* and *RPL15* genes [51].

### 3.5. Prediction of miRNAs Targeting Hub Genes

A total of 17 microRNAs (miRNAs) associated with the hub genes were identified using the miRNet tool, a powerful and comprehensive online tool that integrates more than 14 miRNA databases. The detailed results are presented in Table 1. Several target genes, including *EIF4A1*, *HNRNPA1* and *COX5A*, were found to be associated with three or more miRNAs. Finally, an exhaustive bibliographic review was conducted, limited to studies published in the last five years, to determine if research had been conducted on each of the microRNAs (miRNAs) related to amyotrophic lateral sclerosis (ALS). It was found that some of them, such as hsa-mir-100-5p, hsa-mir-125b-5p, hsa-mir-133a-3p, hsa-miR-206 and hsa-miR-133b [52,53,54,55,56], are closely associated with ALS. On the other hand, certain miRNAs, such as hsa-let-7a-5p and hsa-mir-26a-5p, have been linked to neurological diseases other than ALS [57].

## 4. Discussion

ALS, a devastating neurological disorder, was initially recognized in the 19th century; however, the fundamental etiology and pathophysiological mechanisms of the disease remain elusive to this day [58]. Therefore, importance needs to be given to studying the progression of ALS and developing new therapeutic strategies. ALS is a complex multifactorial pathophysiology in which numerous molecular and cellular processes appear to cause the neurodegeneration of ALS [59].

Currently, many computational approaches and high-throughput multi-omics technologies have been used for the identification of genes and pathways associated with ALS [31]. Remarkably, the study of multi-omics data (transcriptomics, proteomics and metabolomics) that aims to investigate new potential biomarkers and therapeutic targets is becoming key in the study of ALS disease. The original authors of this database, Bernadini et al. [34], performed a DEG study, as well as a functional study, a principal component analysis (PCA) and interconnected biological networks. Therefore, our aim was to introduce a novel approach to use this database with a less restrictive cutoff for DEG analysis to perform functional and enrichment analyses of DEG pathways. We also introduced additional protein–protein interaction (PPI) studies to identify subnetworks and hub genes. These hub genes were then used to predict associated miRNAs.

In this study, after analyzing the gene expression profiles of the patients, we obtained 320 up-regulated genes and 77 down-regulated genes. Compared to the results presented by the original authors [34], our results aligned with the differences in methodology used. We applied cut-off criteria of |logFC| > 0.5 for up-regulated genes and |logFC| < 0.5 for down-regulated genes, whereas cut-off criteria of |logFC| > 1 and |logFC| < 1, respectively, were previously used [34]. These variations in cut-off criteria contributed to the observed differences in the differentially expressed genes that were identified. The most significant pathways identified were related to the development of muscle structure for the up-regulated genes and to the generation of precursor metabolites and energy for the down-regulated genes. These findings were in accordance with the skeletal muscle dysfunction that can contribute to progressive muscle weakness in ALS [60]. In addition, it has also been observed that energy metabolism is altered in human ALS muscle cells, which correlates with energy metabolism pathways [61], as was also suggested by Bernadini and coworkers, indicating that the biological processes were mainly related to skeletal muscle development/contraction and the generation of precursor metabolites and energy [34].

From this group of both up- and down-regulated genes, those genes with the greatest potential as biomarkers were identified. The relevant up-regulated hub genes were *EEF1A1*, *RPLP0*, *EEF2*, *EIF4A1*, *CCT2*, *HNRNPR*, *RPL12*, *RPL15*, *HNRNPA1* and *PABPC1*. No ALS-related scientific evidence was found for the *RPLP0*, *EEF2* and *CCT2* genes. However, some studies related to the *EIF4A1*, *HNRNPR*, *HNRNPA1* and *PABPC1* genes were considered. *EIF4A1* has been found to play a role in the formation of stress granules in motor neurons [44,62]. Elevated expression of *HNRNPR* and *HNRNPA1* together with a subset of human RNA-binding proteins that bind to the GGGGCC repeat RNA of the *C9orf72* gene, one of the most common causes of ALS, reduces the level of GGGGCC repeat RNA, leading to the suppression of neurodegeneration. In addition, the involvement of *HNRNPA1* in the different molecular pathways related to ALS neurodegeneration is increasingly being studied, as this protein is thought to play a key role in mRNA transcription, splicing, stability, transport and translation [46,63]. The *PABPC1* gene has been associated with the *UBQLN2* gene, one of the key genes linked to amyotrophic lateral sclerosis (ALS). Specifically, it has been observed to play a role in regulating stress granule dynamics and the pathogenesis of ALS [47]. All these genes could be potential targets for conducting new studies on their gene expression in ALS and investigating the possibility of them serving as new and potential biomarkers. Finally, although the *EEF1A1*, *RPL12* and *RPL15* genes have not been directly linked to ALS, they have been associated with other neurological diseases, such as Parkinson’s and Alzheimer’s [49,51]. *EEF1A1* has demonstrated involvement in the regulation of genes associated with the neuroinflammatory process in Parkinson’s disease [49]. Conversely, the *RPL12* and *RPL15* genes exhibited significant up-regulation in brain capillary samples obtained from patients diagnosed with Alzheimer’s disease [51]. Notably, these processes have also been described in ALS, especially those ones related to neuroinflammation [64,65], identifying them as interesting candidates for further investigation.

Among the genes that were notably down-regulated (*NDUFA4*, *COX5B*, *COX6C*, *NDUFA3*, *NDUFB4*, *COX5A*, *COX6A2*, *NDUFA6*, *COX8A* and *ATP5MC1*), only *NDUFA4* has been previously identified in connection with ALS. The *NDUFA4* gene is associated with *REEP1* and plays a crucial role in maintaining the integrity of mitochondrial complex IV. It has been observed that the interaction between *NDUFA4* and *REEP1* could block the access of mitochondrial proteases to the proteolysis sites of *NDUFA4* [48]. This gene could be an interesting candidate for study since one of the characteristic pathologies of ALS is the alteration of mitochondrial bioenergetics. Therefore, it would be interesting to conduct future studies to explore its role in ALS. Additionally, many studies have already demonstrated the feasibility of using small molecules to enhance OXPHOS mitochondrial activity as a novel therapeutic approach in ALS [66]. For the rest of the genes, not much information is available, but it is known that they are involved in electron transport processes and ATP syntheses that have been found altered in the ALS pathogenesis [67].

Nowadays, miRNAs provide new avenues for research on diseases. In this study, combining computational and bioinformatics analysis, a total of 17 miRNAs targeting hub genes were identified (12 miRNAs for up-regulated genes and 5 miRNAs for down-regulated genes).

In particular, hsa-mir-100-5p has been associated with neuronal apoptosis in the central nervous system, contributing to the neurodegeneration of motor neurons [52]. One role of hsa-mir-125b-5p is to regulate genes related to DNA repair in ALS associated with the *FUS* gene [53]. Regarding hsa-mir-133a-3p, it has been proposed as a potential preclinical progression biomarker for ALS associated with *G376D-TARDBP* [55]. Studies have also suggested hsa-miR-206 and hsa-miR-133b as promising biomarkers for ALS [54,56,68,69].

In relation to the rest of the miRNAs, relevant information has been found related to other neurological diseases. For example, hsa-let-7a-5p and hsa-mir-26a-5p have been linked to major depressive disorder [57], and hsa-mir-140-3p has been studied for its potential role as a diagnostic biomarker for patients with acute ischemic stroke [70]. In general, this information is valuable to consider, as there is scientific evidence related to this group of miRNAs. This opens the possibility of new study approaches by combining information on hub genes and their miRNAs, providing new ideas for potential targets of studying their role in ALS, such as in the case of hsa-miR-206. Hsa-miR-206 is a miRNA targeting the *HNRNPA1* gene, which, as mentioned earlier, is likely involved in the pathogenesis of ALS. Additionally, this gene is associated not only with hsa-miR-206 but also with hsa-mir-140-3p and hsa-mir-27a-3p. Therefore, these two miRNAs could likely play a role in ALS.

## 5. Conclusions

This study allowed us the opportunity to initiate an exploratory investigation using skeletal muscle samples to identify potential hub gene and microRNAs (miRNAs) of interest and to analyze their roles in amyotrophic lateral sclerosis (ALS) by using bioinformatics tools. These computational tools provide the potential for identifying distinct molecular targets that may exhibit interrelated associations by using data from patients’ samples. Consequently, we suggested some candidate genes for further scrutiny in future investigations. This study presented an initial assessment through bioinformatics analyses. A single gene expression profile from the skeletal muscle samples of sporadic patients was used to provide new insights on this affected tissue. Therefore, we suggest that the information obtained in this study can be validated through cellular experiments, high-throughput analyses, RT-qPCR, next-generation sequencing analyses, animal studies and an even larger cohort. These validations will provide a better understanding of ALS genetics or genetic variations among ALS patients, thereby enabling the monitoring of patients in the clinical practice.

## Figures and Tables

**Figure 1 biomolecules-14-00377-f001:**
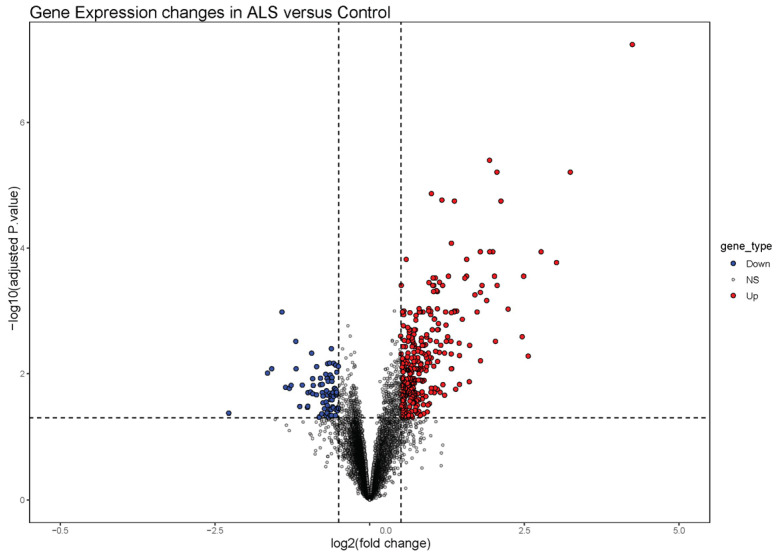
Volcano plot of differentially expressed genes (DEGs). Red dots represent up-regulated genes according to *p*-values < 0.05 and |logFC| > 0.5. Blue dots represent down-regulated genes according to *p*-values < 0.05 and |logFC| < 0.5.

**Figure 2 biomolecules-14-00377-f002:**
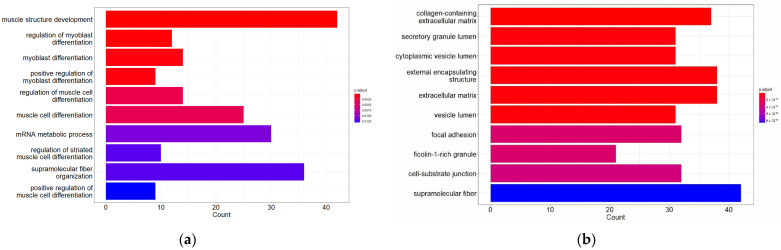
Gene Ontology (GO) function and pathway enrichment analysis of up-regulated DEGs and down-regulated DEGs. The top 10 terms for each of the GO analysis categories’ biological processes (BPs), cellular component (CC) and molecular function (MF) are presented. (**a**) BPs of up-regulated DEGs; (**b**) CC of up-regulated DEGs; (**c**) MF of up-regulated DEGs; (**d**) BPs of down-regulated DEGs; (**e**) CC of down-regulated DEGs; and (**f**) MF of down-regulated DEGs.

**Figure 3 biomolecules-14-00377-f003:**
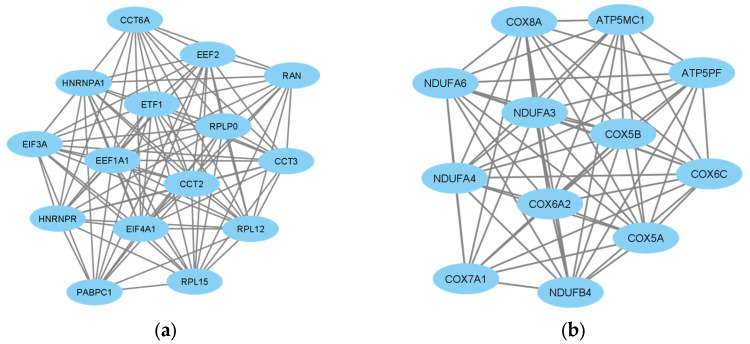
Analysis of differentially expressed gene (DEG) networks. (**a**) MCODE-clustered subnetwork of up-regulated DEGs. (**b**) MCODE-clustered subnetwork of down-regulated DEGs.

**Figure 4 biomolecules-14-00377-f004:**
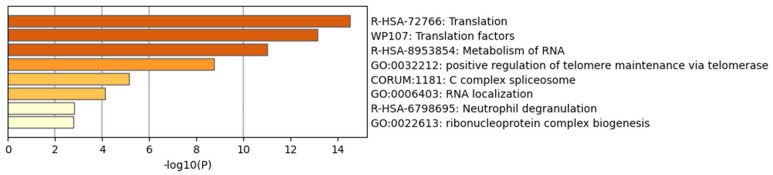
Enrichment analysis of MCODE-clustered subnetwork of up-regulated DEGs by Metascape.

**Figure 5 biomolecules-14-00377-f005:**
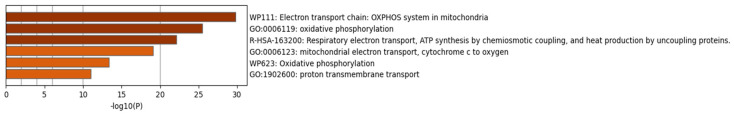
Enrichment analysis of MCODE-clustered subnetwork of down-regulated DEGs by Metascape.

**Figure 6 biomolecules-14-00377-f006:**
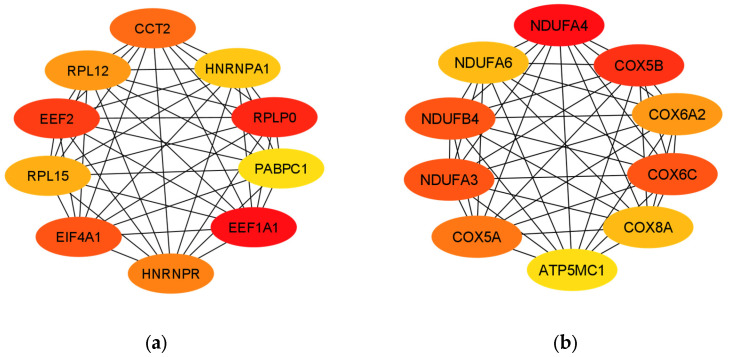
Hub genes identified by cytoHubba. (**a**) Hub genes of the PPI network of up-regulated DEGs. (**b**) Hub genes of the PPI network of down-regulated DEGs.

**Table 1 biomolecules-14-00377-t001:** List of miRNAs and their target genes. Each miRNA is associated with one or more target genes.

miRNA	miRNA Accession	Gene Target	Tissue	Up/Down
hsa-let-7a-5p	MIMAT0000062	*EEF2*	Muscle	Up
hsa-mir-100-5p	MIMAT0000098	*EEF1A1* *RPL15* *RPL12*	Muscle	Up
		*EEF1A1* *RPL15* *RPL12*		
hsa-mir-125b-5p	MIMAT0000423	*EEF1A1*	Muscle	Up
		*RPLP0*		
		*PABPC1*		
		*EEF2*		
hsa-mir-133a-3p	MIMAT0000427	*EIF4A1*	Muscle	Up
hsa-mir-133b	MIMAT0000770	*EIF4A1*	Muscle	Up
hsa-mir-134-3p	MIMAT0026481	*RPL12*	Muscle	Up
hsa-mir-1-3p	MIMAT0000416	*HNRNPA1* *HNRNPR* *EEF1A1* *EIF4A1* *RPL15* *RPL12*	Muscle	Up
hsa-mir-140-3p	MIMAT0004597	*RPLP0 HNRNPA1* *EIF4A1* *EEF2* *RPL15*	Muscle	Up
hsa-mir-193a-5p	MIMAT0004614	*PABPC1*	Muscle	Up
hsa-mir-206	MIMAT0000462	*HNRNPA1*	Muscle	Up
hsa-mir-26a-5p	MIMAT0000082	*EEF2*	Muscle	Up
hsa-mir-27a-3p	MIMAT0000084	*PABPC1 HNRNPA1* *EEF2*	Muscle	Up
hsa-let-7a-5p	MIMAT0000062	*NDUFB4 NDUFA3* *COX8A*	Muscle	Down
hsa-mir-100-5p	MIMAT0000098	*COX5A*	Muscle	Down
hsa-mir-134-5p	MIMAT0000447	*COX8A*	Muscle	Down
hsa-mir-1-3p	MIMAT0000416	*NDUFB4 COX6C* *NDUFA3 COX5A* *NDUFA6*	Muscle	Down
hsa-mir-26a-5p	MIMAT0000082	*COX5A* *COX5B* *COX8A*	Muscle	Down

## Data Availability

Data used in this study are publicly available on the gene expression omnibus (GEO) database and can be accessed through GSE41414.

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
