# Peer review of "Sporadic Amyotrophic Lateral Sclerosis Skeletal Muscle Transcriptome Analysis: A Comprehensive Examination of Differentially Expressed Genes"

_biomolecules, 2024, doi:10.3390/biom14030377_

Round 1

Reviewer 1 Report

Comments and Suggestions for Authors

This is a very interesting examination of altered muscle transcriptome in sporadic ALS. Although sALS represents the vast majority of cases I think that the authors should have discussed their findings in relation to fALS. Similarly, maybe it would make sense to add sporadic to the title to avoid any confusion as the ALS cases included in this study were all sALS. Lastly, there are a few sentences in the discussion that make no sense. I have spotted l245 and l290 but the manuscript should be re-read to make sure there are no surplus words or words missing. Overall I do not have any issues with its publication and I think this would be an interesting paper for the ALS community.

Author Response

Dear Reviewer 1,

Thank you very much for your suggestions and comments. Following your recommendations, we have described point-by-point the changes that have been made in the revised manuscript and they are highlighted in blue.

Our responses to each of the comments are shown below.

  • It would make sense to add sporadic to the title to avoid any confusion as the ALS cases included in this study were all sALS.

 We you are very grateful for this comment. We agree that in this study all cases are sporadic, and therefore we have modified the title to better fit with the main focus of the study. Additionally, we adjusted the abstract and text to be more specific to sporadic ALS.

  • Lastly, there are a few sentences in the discussion that make no sense. I have spotted l245 and l290 but the manuscript should be re-read to make sure there are no surplus words or words missing.

We also thank you for this comment. We have revised and corrected the sentences in lines L251-L253 and L312-L314, as well as the whole manuscript to ensure that all the sentences are correctly written.

Kind regards,

Elisa Gascón

Reviewer 2 Report

Comments and Suggestions for Authors

In this study, Osta and colleagues analyzed the publicly available microarray data to identify potential biomarkers and therapeutic targets for amyotrophic lateral sclerosis (ALS) by examining differentially expressed genes in skeletal muscle tissue. The authors have conducted a comprehensive analysis including functional enrichment, protein-protein interaction network analysis, and miRNA target gene network analysis. The findings provide a useful resource for future experimental validation studies related to ALS pathogenesis.

Major Comments:

1. The authors utilized a database which was previously investigated and published (PMID: 23469062).  This prior study had examined the mitochondrial network genes in the skeletal muscle of amyotrophic lateral sclerosis patients. However, the authors failed to acknowledge this prior study in their own research. The authors should compare their analysis with the findings of the previous study to highlight any discrepancies and present any novel insights resulting from their new analysis.

2. The authors conducted a differential expression gene (DEG) analysis using a fold-change cutoff of |logFC| > 1.42 for up-regulated genes and |logFC| < 0.707 for down-regulated genes, which deviates from typical values. I'm curious about the rationale behind this choice and whether there is any literature supporting these criteria.

3. The authors should provide more justification for focusing solely on skeletal muscle tissue, as ALS primarily affects motor neurons. While muscle atrophy is a key symptom, analyzing motor neuron data could potentially reveal more direct insights into disease mechanisms.

Minor Comments:

1. In the introduction, the authors should update the statement "Presently, only two approved drugs, edaravone and riluzole, are available for ALS treatment," as a third drug, Sodium Phenylbutyrate and Taurursodiol, is now approved by the FDA.

Author Response

Dear Reviewer 2,

Thank you very much for your suggestions and comments. Following your recommendations, we have described point-by-point the changes that have been made in the revised manuscript and they are highlighted in blue.

Our responses to each of the comments are shown below.

  • The authors utilized a database which was previously investigated and published (PMID: 23469062). This prior study had examined the mitochondrial network genes in the skeletal muscle of amyotrophic lateral sclerosis patients. However, the authors failed to acknowledge this prior study in their own research. The authors should compare their analysis with the findings of the previous study to highlight any discrepancies and present any novel insights resulting from their new analysis.

We completely agree with this comment and we have modified the study by quoting (Quote 33) and adding the information from the previous study entitled: "Mitochondrial network genes in the skeletal muscle of amyotrophic lateral sclerosis patients" by Bernardini C. and coworkers. Interestingly, we used a less restrictive |logFC| cuttoff, so we obtained a higher number of up- and down-regulated genes compared to the original authors (L261-L276), as expected. Finally, we added a brief comparison of the results obtained in the functional enrichment (L270-L284). The findings we provide are new additional studies not conducted by the original authors as far as we know.

  • The authors conducted a differential expression gene (DEG) analysis using a fold-change cutoff of |logFC| > 1.42 for up-regulated genes and |logFC| < 0.707 for down-regulated genes, which deviates from typical values. I'm curious about the rationale behind this choice and whether there is any literature supporting these criteria.

We are very grateful for this comment. We apologize for not having explained the value of logFC in more detail. To clarify, the selected values of |logFC| > 1.42 for up-regulated genes and |logFC| < 0.707 for down-regulated genes represent log2-transformed values equivalent to |logFC| > 0.5 and |logFC| < 0.5, respectively. In simpler terms, log2(0.5) equals 1.42, and log2(-0.5) equals 0.707. We have adjusted the initial values in the manuscript (L91-L92 and L142-L143) to better clarify this cutoff. Using the new criteria, we obtained 320 up-regulated genes and 77 down-regulated genes.

The rationale behind choosing a 1.4-fold change for up-regulated genes and a 0.7-fold change for down-regulated genes is to enhance the inclusivity of genes in our study. This decision deviates from the more restrictive cutoffs employed by the original authors of the database, who used |logFC| > 1 and |logFC| < 1, resulting in 80 up-regulated genes and 16 down-regulated genes ("Mitochondrial Network Genes in the Skeletal Muscle of Amyotrophic Lateral Sclerosis Patients" by Bernardini C. and coworkers).

Interestingly, similar cutoffs have been used in many studies. For instance, the studies by Tiefeng Cao and coworkers ("Identification of fatty acid signature to predict prognosis and guide clinical therapy in patients with ovarian cancer", doi: 10.3389/fonc.2022.979565),  Zheng He and coworkers ("Crosstalk between Venous Thromboembolism and Periodontal Diseases: A Bioinformatics Analysis", doi: 10.1155/2021/1776567) and Wentao Yang and coworkers ("ABSSeq: a new RNA-Seq analysis method based on modeling absolute expression differences", doi: 10.1186/s12864-016-2848-2) used the same cutoff as the one we have used in this study.

  • The authors should provide more justification for focusing solely on skeletal muscle tissue, as ALS primarily affects motor neurons. While muscle atrophy is a key symptom, analyzing motor neuron data could potentially reveal more direct insights into disease mechanisms.

We agree with this comment and we also think the analysis of motor neuron data could provide more direct insights into the mechanisms of the disease. In fact, numerous studies have been already based on motor neuron data. As an example, “Key Molecules and Pathways Underlying Sporadic Amyotrophic Lateral Sclerosis: Integrated Analysis on Gene Expression Profiles of Motor Neurons” by Lin J. and coworkers (doi: 10.3389/fgene.2020.578143).

Our main aim was to provide new insights using a tissue that is also mainly affected by ALS disease and becomes progressively atrophied during disease progression since it is in clear connection with motor neuron. For this reason, our study was focused on skeletal muscle. In addition, a growing number of studies highlight that skeletal muscle undergoes pathological changes from the early stages of the disease such as the studies suggested by Verma S and coworkers (“Functional consequences of familial ALS-associated SOD1L84F in neuronal and muscle cells”, doi: 10.1096/fj.202301979R) and Ferraiuolo L and coworkers (“Transcriptional response of the neuromuscular system to exercise training and potential implications for ALS”, doi: 10.1111/j.1471-4159.2009.06080.x). We have clarified this point in the Introduction section on page 2 (L54-L57).

  • In the introduction, the authors should update the statement "Presently, only two approved drugs, edaravone and riluzole, are available for ALS treatment," as a third drug, Sodium Phenylbutyrate and Taurursodiol, is now approved by the FDA.

We thank you for this comment. Sodium Phenylbutyrate and Taurursodiol received FDA approval for ALS treatment. The comprehensive study for this treatment was conducted by Amylyx, designated as AMX0035. Unfortunately, this study ended very recently and could not finally provide effective results. Please, find attached the related document to this study.

Kind regards,

Elisa Gascón

Round 2

Reviewer 2 Report

Comments and Suggestions for Authors

The authors have adequatley addressed my questions.